# Blending Behavioural Theory and Narrative Analysis to Explore the Lived Experience of Obesity and Assess Potential Engagement in a UK Weight Management Service: Theory and Narrative Approaches in Weight Management

**DOI:** 10.3390/healthcare12070747

**Published:** 2024-03-29

**Authors:** Jessica Gillespie, Hannah Wright, Jonathan Pinkney, Helen Lloyd

**Affiliations:** 1School of Psychology, University of Plymouth, Plymouth PL4 8AA, UK; jessgillespie@hotmail.co.uk (J.G.); hannah.wright2798@gmail.com (H.W.); 2Peninsula Medical School, University of Plymouth, Plymouth PL4 8AA, UK; jonathan.pinkney@plymouth.ac.uk

**Keywords:** obesity, weight management, narrative elicitation, narrative-based medicine, motivation, locus of control, self-efficacy, attribution style, stages of change, transtheoretical model

## Abstract

Background: Current treatments for people with obesity emphasise the need for person-centred approaches that consider complex biopsychosocial factors and value the lived experience of people when attempting to lose weight. Methods: Narrative interviews (*n* = 20) were conducted with people living with obesity to explore the causes of their weight gain and their expectations and engagement with treatment at a Weight Management Clinic. A mixed inductive and deductive qualitative analysis identified utterances that represented psychological constructs used to understand self-appraisal and health behaviour. A narrative analysis was used to situate these findings in the context of a participant’s life story. Results: Locus of control was a dominant construct evidenced through a person’s attributional style and self-efficacy. Transcripts represented a heightened sense of self-understanding and shifts in control, and styles of attribution and efficacy resulted in either stasis or self-actualisation. The Stages of Change model could be applied to narratives to ascertain a patient’s motivation to access treatment. Importantly, narrative interviews also allowed for the consideration of how a person’s systemic context influenced their weight. Conclusion: Narrative interaction supports both self- and shared understandings of the causes and consequences of obesity for individuals, in a non-blaming or shaming manner. It provides an opportunity to enhance engagement through tailored, person-centred treatments.

## 1. Introduction

Obesity has become a major global health problem and is linked to various co-morbidities [1]. The impact of obesity on the physical health of sufferers and the economic burden it exerts on health services are often widely reported, without active consideration of the individual and their systemic context [2]. Tackling obesity has received renewed interest since the COVID-19 pandemic, with studies suggesting that people living with obesity were more likely to experience extreme symptoms and mortality [3,4]. As a result, the UK government announced public health measures such as banning the advertising of junk food [5]. In recognition of the wider environmental and psychosocial drivers of obesity, current research and national and international guidelines are emphasising the need for person-centred approaches to the detection, screening, treatment, and management of obesity [6,7,8,9]. This signals a departure from reductionist perspectives that attribute causality to conscious lifestyle choices easily remedied by ‘willpower’ and healthy eating and exercise [10]. Despite this, the relationship between obesity, mental health, social inequality, and material deprivation are often unheeded. A lower socioeconomic status has been found to predict a higher Body Mass Index (BMI), implicating a range of important social mechanisms such as employment, family status, education, and housing [11]. In addition, more women than men suffer from obesity and they are more likely to seek treatment [12]. Furthermore, the latest figures from the UK show that people who identify as black, live in deprived areas and have lower levels of education are also more likely to suffer from obesity [12]. With such strong social determinants, it is argued that obesity should be tackled by both economic and social interventions [13].

The psychological aspect of living with obesity Is often overlooked. Being overweight or obese can be both the cause and consequence of mental health difficulties [14], with individuals facing discrimination and increased levels of depression, anxiety, and stress [14]. Health policies designed to tackle obesity have ranged from nutritional standards in schools to the creation of specialised weight management programs, but few interventions have had the desired impact at a population level [9]. There is a clear need for more holistic approaches that take into account complex biopsychosocial influences.

Various psychological theories have been proposed to help elucidate the complex nature of obesity, from development through to maintenance and treatment. Behavioural theory suggests that learned behaviours and habits related to eating, physical activity, and lifestyle choices are key [15]. Behavioural theories also consider the role of environmental factors, such as food availability, social influences, and cultural norms, in shaping eating behaviours. Such theories propose that obesity results from the excessive consumption of high-calorie foods, a sedentary lifestyle, and a person finding it difficult to regulate their eating behaviours [16]. Cognitive theorists, on the other hand, prioritise the impact of thoughts, beliefs, attitudes, and perceptions on eating behaviours and weight regulation [17]. These theories suggest that distorted cognitions about body image, food, and weight contribute to overeating and weight gain [17].

The roles of stress, distress and social interactions are considered integral in psychosocial theories about obesity and are considered by the British Psychological Society (BPS) as applicable within existing models [18]. The BPS endorses biopsychosocial theory as the most informative framework for understanding obesity. This model integrates biological, psychological, and social factors to understand the multifaceted nature of obesity. It incorporates the impact of social determinants, genetic predispositions, neurobiological mechanisms (e.g., reward pathways, hormonal regulation), psychological vulnerabilities, and environmental influences on the development of obesity and obesity-related behaviours [18]. This theory can account for stress eating, emotional eating, and using food as a coping mechanism within a person’s social context and interpersonal relationships, which are behaviours often neglected within other theories [18].

The application of psychological theory has the potential to explain health-related behaviour and, specifically, provide a clinical understanding of a person’s ability to engage with a weight management treatment program. However, the application of psychological theory to complex biopsychosocial conditions can be difficult, and it often lacks ecological and contextual validity. From a candidate list of 82 behavioural theories [19], we subjected four theories to analytical scrutiny against an ecologically situated dataset derived from narrative interviews regarding the lived experience of obesity. Theories were selected that had the potential to provide explanatory power within a situated psychosocial framework, but that importantly acknowledged relationships between the self, an individual’s immediate social relations, and the wider facilitators and constraints on behaviour and experience over time. The selected theories provided a robust integration of the behavioural, cognitive, and social constructs used in psychological theories of obesity and, importantly, held explanatory potential as a combined model of engagement with the treatment offered at a Weight Management Clinic (WMC). The rationale for the selection of each theory is provided below.

Attribution theory [20] was selected because it describes the process by which individuals assign causes to events (e.g., whether internal or external), and has been shown to impact the choice and efficacy of treatment in diabetes [21]. Locus of control (LoC), or the extent to which people believe events in their lives are within their control, has been shown to influence engagement with behaviour change aimed at reducing cardiovascular disease risk [22]. Attribution and control are also likely to relate to a sense of self-belief regarding the achievement of attainment goals; this is known as self-efficacy [23], and has been linked to improved outcomes for eating disorder treatment [24]. 

The Stages of Change model (SoC), otherwise known as the Transtheoretical model [25], proposes that individuals progress through five behavioural stages: Precontemplation, Contemplation, Preparation, Action, and Maintenance [26]. The precontemplation stage describes no intention to change behaviour within the foreseeable future. Contemplation describes awareness of the problem but that the person is yet to begin any action. Preparation is when individuals are making small changes but have not taken effective action. Action is when individuals modify appropriate factors to overcome problems. Finally, the maintenance stage is when individuals work to prevent relapse. The SoC framework has been used in weight management services to assess the potential of individuals to engage and subsequently tailor interventions in accordance with their readiness [25]. A SoC assessment is often derived by administering a standardised questionnaire [26]. This reductionist approach does not capture the complexity of factors shaping an individual’s ability to engage in treatment [27]. A more contextually situated approach is needed to optimise treatment and improve outcomes [28,29].

Narrative-based medicine may be the key to a more person-centred understanding of the factors that impact a person’s weight gain and motivation to lose weight [30]. Narratives provide a perspective of a person in an idiosyncratic context with individual needs, rather than a diagnosis with a biological cause, as the biomedical model would suggest [31,32]. This is especially important for our emerging conceptualisation of obesity as complex and multifactorial. Enhancing our understanding of the internal and external impacts an individual experiences due to health conditions or illnesses can increase our understanding of their ability to engage in relevant interventions [32]. Higher levels of engagement in health-related interventions are associated with positive therapeutic relationships [33]. A narrative approach has been proposed specifically to strengthen patient–professional relationships, which can in turn improve health outcomes [31]. Additionally, narratives can enhance individuals’ motivation for change by dialectically surfacing barriers and facilitators, bringing them into useful dialogue with each other [34,35] and thereby reducing ambivalence towards change.

This study analysed narrative interviews conducted with people referred to a UK WMC. It aimed to test the applicability of psychological theories of behaviour empirically grounded in people’s lived experiences in order to elucidate the biopsychosocial factors impacting a person’s ability to engage with treatment.

## 2. Materials and Methods

### 2.1. Design

Twenty people referred to a UK WMC gave written, informed consent to take part in a narrative interview for this study. Interviews were transcribed verbatim and subject to a combined theoretical and narrative analysis. The interpretive approach was underpinned by ontological relativism, which acknowledges that people create their individual realities based on their social experiences in the world, and epistemological constructionism, which suggests that knowledge is both co-constructed and subjective [36].

### 2.2. Role of the Researcher(s)

The researcher who carried out the interviews was a doctoral student with a psychology background and no prior knowledge of obesity or weight management clinic settings. AUTHOR 1, who analysed the transcripts and drafted the manuscript, was studying clinical psychology and had an interest in clinically applied interventions. AUTHOR 2 analysed the transcripts and helped to draft the manuscript. AUTHOR 3 held a lead role in the WMC and provided a clinical perspective on the findings and the manuscript. AUTHOR 4 devised the study, co-wrote the manuscript, dual coded transcripts and discussed emergent findings with AUTHOR 1. The researchers involved in data collection and analysis kept reflexive diaries and actively challenged emerging assumptions during analysis.

### 2.3. Participants and Procedure

In order to recruit a consecutive convenience sample of 20 eligible participants, patients who were newly referred and about to start treatment and those that were within early treatment at the WMC (e.g., within the first nine weeks of treatment) were approached. The treatment timeframes vary for tier 3 WMCs, but typically last between 6–9 months before bariatric treatment is considered. Treatment consists of one-to-one sessions, group sessions, a buddy scheme and input from a multidisciplinary team, including a psychologist.

The narrative interviews explored why the participants were referred to the WMC; how people felt about their weight; their expectations of the WMC; and whether they had any weight-related goals. The interviews took place in the participants’ homes and ranged from 36 min to 2 h. All participants identified as White British, they were aged between 20 to 82 years, and 6 were male and 14 were female.

### 2.4. Data Analysis

AUTHOR 1 read the transcripts and crosschecked these with the audio files. The transcripts were imported into and coded using nVivo v12 software. A deductive theoretical analysis explored the text within each transcript and across all transcripts [37] to identify utterances relating to Attributional style (AS), Locus of Control (LoC), and Self-efficacy (SE) in relation to motivation to lose weight. When no new codes emerged, existing codes were collated for refinement and mapping onto the theoretical framework. AUTHOR 1 applied Labov’s (1972) [38] structural analysis (see Appendix A
Table A1, which explains Labovian codes) to investigate the patterning of stories and the salient content. Participants’ accounts of referral to the WMC were analysed as their ‘referral sub-plot’ and used to further scrutinise text related to AS and LoC. The transcripts were also analysed in their entirety to assess whether they contradicted or supported referral sub-plots. The participants were placed in a SoC category based on the theoretical and structural analyses, according to Norcross et al.’s (2011) [26] definitions. Storylines were visualised to illustrate the shifts in the externalisation and internalisation of LoC, with the starting point being their referral sub-plot.

Throughout the analysis, Wong et al.’s (2018) [39] narrative analytic levels (personal, interpersonal, and social) were utilised. This provided a means of identifying how stories were represented within the interviews, and the role of wider discourses on how the stories were told [39].

### 2.5. Trustworthiness and Rigor

Analyses were undertaken by co-author AUTHOR 4 to validate or challenge AUTHOR 1’s preliminary analysis. Issues were resolved through dialogue and re-checking the data. AUTHOR 3 also checked the transcripts against the emerging theoretical framework to scrutinise the representativeness of the data. AUTHOR 4 used a narrative analysis template blinded to AUTHOR 1’s allocation to cross check the referral sub-plots and the different levels of narrative as they emerged across the transcripts (personal, interpersonal, social). These quality checks ensured that the methodological strategies were robust, credible, and rigorous [36].

## 3. Results

### 3.1. Theoretical Analysis

Text supporting the theoretical constructs (AS, LoC and SE) was identified deductively from the transcripts. Three inductive themes also emerged throughout this analysis, as depicted in the conceptual map (Figure 1): Environment, Upbringing, and Mental Health. These themes helped explain AS, LoC and SE.

#### 3.1.1. Locus of Control

The analysis revealed that LoC formed an over-arching construct underpinned by AS and SE. Attributional talk was related to past events, and self-efficacy talk was related to future events. Discrete patterns of AS and SE talk determined a participant’s appraisal of their control of life events. For example, participants who spoke of uncontrollable life events causing weight gain demonstrated an externalised AS, which often coincided with a perception of low SE. These participants were often in an earlier ‘SoC’, with lower levels of motivation and greater barriers to engaging in weight loss activities. In contrast, participants who acknowledged more control over their life events expressed internalised AS and a higher sense of SE. With higher levels of motivation to lose weight, these individuals were fit to a later SoC. Thirteen of the twenty participants represented this pattern.

#### 3.1.2. Attribution Style

AS was commonly signalled through a narration of the events in the past that led to referral to the WMC or the reason for weight gain. All participants clearly conceptualised what initiated their weight difficulties, which were mostly attributed to external events. This is perhaps a protective belief, since locating the cause of weight gain as outside of one’s control is destigmatising. However, this negatively impacted the individuals’ motivation to lose weight since it resulted in less perceived control over external causative factors. Participants who attributed their referral internally appeared more motivated to engage and resided in a later SoC, compared with those who attributed their referral to the will of a healthcare professional.

#### 3.1.3. Self-Efficacy

SE was talked about in relation to future actions. Differences in SE across participants were related to where a person resided in the ‘SoC’ model; talk suggesting higher SE was more commonly associated with ‘preparation’ or ‘action’, and talk suggesting lower SE related to ‘contemplation’. The way participants spoke about the barriers to engaging with the WMC impacted their self-efficacy and motivation. Practical barriers, such as the cost of transport to attend groups and health barriers such as suffering from chronic pain that inhibited exercise, were commonly discussed. Participants that spoke of barriers without evidencing how to overcome them expressed less motivation and were placed in an earlier ‘SoC’.

#### 3.1.4. Environment

Participants spoke about the impact of their environment on their weight in the present or recent past, with the potential for it to moderate motivation. Feeling unsupported and not listened to by professionals was a common experience across the interviews and was cited as a previous barrier to engaging with weight loss strategies. It was also considered as a potential future barrier to engagement with weight management services. The impact of lifestyle on a person’s weight was evident, and this mostly related to employment, especially in relation to work stress and shift work. Participants reported that negative and stigmatising comments received about their weight influenced how they felt about themselves, reinforcing their lack of confidence to lose weight.

#### 3.1.5. Upbringing

Related to ‘Environment’, this explanatory construct felt qualitatively different because of its temporal location in the distant past. Participants often made links between childhood neglect and eating habits, expressing that they had not been educated about healthy eating. They also expressed that as children they had often been rewarded with sugary or fatty foods, which they felt had subsequently increased their consumption of such products in reward pathways.

#### 3.1.6. Mental Health

Located in talk about the present, this construct described how participants felt their weight had an impact on their day-to-day life. Low self-esteem permeated the participants’ responses and was related to the avoidance of behaviours associated with weight loss, such as exercise. Participants also spoke of avoiding stigmatising situations, e.g., gyms, revealing how societal discourse impacts behaviour at an individual level. Many participants spoke of a loss of identity through weight gain, which directly affected their mental health. Participants described their eating behaviours as tools to regulate emotion, to deal with stress and trauma, or to ‘fill an emptiness’ unmet by interpersonal relationships. Many participants directly conceptualised this as comfort eating.

Table A2, found in the appendix, presents each construct and sub-construct introduced above and the number of transcripts that evidenced them, with a selection of illustrative quotations. The participant ID number is presented in brackets.

### 3.2. Narrative Analysis: Storylines and Stages of Change

During analysis, it became clear that ambivalence and contradiction permeated the stories told, representing common features of narrative accounts [34]. As participants vocalised their stories, they embarked on a journey that explored the reasons for their difficulties, their expectations for treatment and their level of motivation to lose weight. Labov’s (1972) [38] structural analysis helped understand how these stories changed over the course of the interview. In all cases, participants’ motivations shifted, as did their LoC, AS and perceived SE.

According to our analysis, four participants had not yet received treatment at the WMC at the time of interview; two of these individuals were classified in ‘contemplation’, and two were in ‘preparation’. Three individuals were going through induction at the WMC at the time of interview; one of these individuals was classified in ‘action’, with the other two individuals in ‘preparation’. Of the seven individuals in early treatment, five were in ‘contemplation’, and two were in ‘action’. Of the six individuals who had received three or more weeks of treatment, three individuals were in ‘action’, two were in ‘preparation’, and one was in ‘contemplation’. This suggests that the earlier into treatment a participant was, the more likely they were to have an earlier SoC. Figure 2 illustrates this.

The analysis revealed shifts to internally and externally located explanations (see Figure 3, Figure 4 and Figure 5), revealing how elements of a person’s story were used to allocate them to a SoC category. A representative case example for each SoC category is presented below to illustrate these findings.

#### 3.2.1. Stages of Change: Contemplation

Storylines for those in contemplation followed a similar pattern. Plot lines start further towards externalisation compared with those of other stages. They then dip into some form of internalisation, and then steadily rise back up to externalisation. Some seem to be intrinsically motivated shifts to internalisation, such as P8 taking control of her health during pregnancy to ensure a healthy baby. Meanwhile, others are extrinsically motivated, such as P18 attending Weight Watchers because she felt pressured to by others. The current externalisation exhibited, as illustrated by the end point of the storylines, supports their categorisation in ‘contemplation’. Contemplators seemed to attribute their weight gain to a health problem or biological cause (P3, P6, P11), or a stressful life event where weight could not be prioritised (P8, P9, P12). They presented their story in a way that expressed that they had tried all they could to lose weight, with no success. It is important to note that participants may have vocalised that their weight gain was caused by external factors to avoid stigmatisation and blame [40].
*Representative Case for Contemplation Phase.***Jane—‘There’s nothing I can do, I’m an awkward one.’**Jane attributes the cause of her weight gain to being diagnosed with diabetes and prescribed insulin in her 30s, when ‘the weight walloped on’. She describes a long battle since, trying various avenues to weight loss, such as attending groups, medication, and exercise, all of which have failed. She made these attempts with her son who had success when she did not, contributing to her identification as ‘one of the awkward ones’. Jane’s WMC referral was suggested by her GP. She had reservations because she has tried for so long to lose weight. She attended one group and left feeling angry as the practitioner did not listen to her, and as a result did not attend the second. She is clearly still thinking as to whether she will engage with treatment or not, which places her into the contemplation Stage of Change.

#### 3.2.2. Stages of Change: Preparation

Participants in this group oscillated between feeling as though they could ‘take control’ of their weight and feeling overwhelmed by life, with their priorities shifted elsewhere. Those in preparation had more awareness of the changes they needed to make and how to achieve them, compared to those in the contemplation group. All had already made small changes in an attempt to lose weight, which revealed a more internalised LoC, and an understanding of the link between behaviour and weight loss/gain. P5, 15, 16 and 19 all showed awareness of the links between binge and comfort eating as ways to manage emotions, and a desire to change these behaviours as a way to manage weight. P7 exhibits less oscillation but recounts the need to lose weight to help the pain he experiences due to cancer of the spine. This is less representative of the group as a whole, as his line is a slow downwards arc and does not exhibit oscillation.
*Representative Case for Preparation Phase.***John—‘Coping with life events through food.’**John was active growing up and injured his knees playing football aged 14. Weight started to become a problem 15 years ago when he discovered his wife was cheating on him and they separated. A two-year period of using alcohol and food to cope with his emotions took place, until he met his current partner. John reports that they currently drink less alcohol, but recognises that his eating habits are still problematic. He did, however, go several weeks without eating takeaways prior to going on holiday recently. John needs knee surgery but has been told he must lose weight before the operation can take place. The nurse and his partner encouraged him to attend the WMC. His goal is to lose weight so he can get surgery for his knee and return to work, something which is important to him.

#### 3.2.3. Stages of Change: Action

More extreme shifts occur in the action-phase storylines compared to those of the other groups. This perhaps illustrates how participants are in a place where they can reflect on the internalisation/externalisation that has previously happened. A common occurrence in this group was a serious health scare. This could be described as a form of biographical disruption [41], where a major event disturbs the structure of everyday life and forces attention to bodily states not usually thought about. Participants who were informed that their weight was causing health problems (P4, 10, 17, 20) had to rethink their sense of self from a ‘healthy’ to ‘unhealthy’ person. Their response to this self-disruption involved a mobilisation of resources, represented by help-seeking and behaviour change. This is illustrated in Joe (see below), who wants to lose weight so he can keep working and return to his hobbies. Participants in ‘action’ expressed more personal responsibility for their weight compared to the other groups, who spent longer externalising and only alluded to self-responsibility briefly. Although health concerns were present in other groups (e.g., P5; preparation—wanting to lose weight for his general health), the narratives of this group suggest that the specific and serious health-related triggers they experienced have prompted a change in behaviour to regain a sense of a ‘healthy self’. Most of this group had commenced treatment, and the decline in their lines at the end towards internalisation could perhaps reflect their engagement with interventions at the WMC.
*Representative Case for Participants in Action***Joe—‘Health scares and taking responsibility’**Joe had a difficult relationship with his father, so lived with his grandparents who displayed love through cooking Joe desserts, giving him large portions, and offerin sweet treats. As a result, he described being a ‘fat kid’. Whilst employed as a refuse collector, he ‘ran the weight off’. However, he then got a driving job and quit smoking, which led to increased eating, alcohol consumption and weight gain. He describes weight impacting his life as he is not as agile when climbing up and down from his lorry and it stops him from working on cars, which he enjoys. Since having serious heart problems, he has bought smaller plates and cutlery to decrease his portion size and ‘trick his mind’ into eating less. He feels he has lost weight as his clothes are loose. Six months ago, Joe told his nurse he would like help with his weight, so she referred him to the WMC.

#### 3.2.4. Synthesis of the Labovian Analysis

The storylines show the complexities in each narrative; each person had a nuanced way of telling their story and many contradictory positions arose at different points during the interviews. The figures show a correlation between how each participant ended their story and the Stage of Change in which they were placed. Through storytelling, they arrived at a particular conclusion in terms of their level of motivation, an important function of narratives [34,42].

## 4. Discussion

This study explored whether patient narratives could be used to assess the utility of psychological theories of behaviour in the context of motivation to engage with a WMC. The theoretical map (Figure 1) illustrated several complex relationships. The constructs supported narrative ideas, as each was temporally located, perhaps reflecting how participants were interviewed at a juncture in their lives, at point of referral to the WMC, or early in treatment. They were required to look to the past to explore the reasons for their weight difficulties, to the present to explore how their weight currently impacts them, and to the future to explore expectations of treatment. The Labovian (1972) [38] structural analysis indicated that information initially presented as salient, within referral sub-plots, changed through the course of the interviews. Shifts in LoC, AS and SE occurred through storytelling. The storylines showed the patterns that existed across the Stage of Change groups; levels of internalisation or externalisation correlated with how motivated the patients were to change.

Using Wong et al.’s (2018) [39] narrative analytic device as a lens via which to view the data, it is suggested that internalised negative social discourses lead patients to attribute their weight to external factors, thus presenting a more socially acceptable narrative to the interviewer and avoiding stigmatisation. The analysis illustrated that LoC is represented through AS and perceptions of SE. These, in turn, signal levels of motivation to change behaviour, and how these are supported or thwarted by a person’s social environment, relationships, socioeconomic status, upbringing, and stigma. Although complex and idiosyncratic, the way in which participants told their stories signalled how ready they were to embark on treatment.

Giving patients time to vocalise their narratives was an empowering strategy that helped them to make sense of the causes and consequences of their weight [39]. Pawson (1996) [42] proposed that interviews should involve the researcher and subject theorising together, towards a shared understanding of complex issues. Exploring their personal stories with an empathic listener helped them construct a coherent explanation and causal model of their journey towards a change, irrespective of how far down the road they had travelled, and provided a map to chart the journey; this is a process of therapeutic employment, as documented by Mattingly (1998) [43]. This narrative approach highlighted the utility of a patient and professional partnership [44]. This could help services to see patients as human beings with histories and goals when assessing motivation and engagement [45], representing partnership to plot the story and chart the terrain.

It became clear that many participants had experienced traumatic life events that impacted their ability to prioritise themselves, their health, and their weight. Participants faced relationship breakdowns, single parenthood, financial struggles, abuse, and bereavements. The externalisation depicted in their storylines does not represent the idea that they were not trying to lose weight, but that for periods of time it was difficult to prioritise. Services need to be aware of this when supporting patients, to ensure that they are not inadvertently blaming individuals or locating problems within people [46]. Trauma-informed approaches and interventions are suggested to be appropriate for those struggling with obesity, contributing to the idea that the ‘problem’ should be located within the social environment, rather than the person [47]. This emphasises the importance of treating each person as an individual with different life contexts and listening to their story in a person-centred and empowering manner.

Using Wong et al.’s (2018) [39] narrative analytic devices allowed for a consideration of the social narratives at play within the interviews. Traditionally, societal attitudes towards people with obesity have been negative [48], where individuals are blamed and shamed for their weight [40]. These data showed a complex relationship between self-esteem, avoidance, and stigma. Internalised negative stigma led to low self-esteem, which resulted in patients avoiding stigmatising situations that could aid weight loss, such as exercise classes. Validating the prejudice and discrimination that patients face is paramount in helping them manage it and restricting its interference with treatment. Receiving such validation may also be the first experience patients have of healthcare professionals considering how societal factors impact weight loss. For this to happen, health practitioners must themselves be aware of the negative biases they hold around obesity. This idea is highlighted by the concept of epistemic injustice in healthcare, testimonial injustice and in particular [49]. This is where a speaker is deemed to have a ‘credibility deficit’ or is not believed due to a particular characteristic, such as being ‘a fat person’, [50]. Practitioners having the reflective capacity to address their biases would ensure that they are treating people as credible witnesses to their experience and would offset testimonial injustices [49]. Understanding how broader societal narratives affect storytelling along with the professional’s judgement is important when assessing motivation to change. Stigmatisation, prejudice, and inequality pervade clinical communication, and eliciting narratives from patients is a way to address this [51].

This study strengthens the argument for using narrative elicitation in weight management services. It can capture and make sense of the intricate biopsychosocial factors involved in the causes and maintenance of obesity, as well as motivation to lose weight. It allows for a critical perspective of healthcare, including how the professional within the clinical interaction impacts what is being said, how societal beliefs affect the person, and any factors that inhibit them from telling their story. These are crucial things to consider in clinical communication, and thus motivational assessments.

### Strengths, Limitations, and Implications

The application of Labov’s (1972) [38] structural categories allowed for an analysis that went beyond the surface of the text or content of the stories, and the application of Wong et al.’s (2018) [39] narrative analytic devices allowed for an exploration of how social narratives impacted how the stories were told. This rigorous and systematic analysis is a strength of the study.

There are also limitations. The interviews were not conducted by the authors, making it difficult to reflect on how the interviewer impacted the construction of the narratives [52], a particular problem when examining Wong et al.’s (2018) [39] ‘interpersonal story’ level. Participants may have presented different versions of their stories to the female researchers compared to the male researcher present at the time. Fourteen of the participants were female, so they may have been more comfortable discussing personal information with those of the same gender. Of course, this is speculation and cannot be validated with respondents due to the interviews having been conducted and anonymized in 2015/16. Not being involved in data construction meant that the authors had to ensure familiarity with the data, which occurred via reading the transcripts and listening to the audio repeatedly. This sample of participants reflect the wider patterning of obesity statistics in relation to prevalence and help-seeking with regard to gender and deprivation, but not with regard to ethnicity. This is due to the sociodemographic profile of the largely white British population of the area.

## 5. Conclusions

This study demonstrated that narrative elicitation provides an opportunity to gain a contextualised understanding of patient experiences, social circumstances, and motivation to engage with weight management treatment. In doing so, it provides knowledge that can be used to support individuals in a targeted manner to improve their weight loss outcomes, whether by behavioural modification or by enhanced knowledge of the personal, social, and environmental factors that prohibit weight loss. The analysis allowed for a critical perspective on the interpersonal and societal factors that influence people who struggle with weight. We hope that this will allow for the more compassionate and empathic treatment of patients referred to weight management services.

Future research could test the SoC framework to inform individualised pathways to treatment, e.g., motivational interviewing for early SoC [53] compared to group-based interventions for those in a later SoC. Building on trauma-informed approaches would also help to offset the shame and blame that people living with obesity feel [47]. The potential for narrative approaches within obesity treatment should be tested in relation to its potential impact on self-efficacy, motivation, and engagement.

## Figures and Tables

**Figure 1 healthcare-12-00747-f001:**
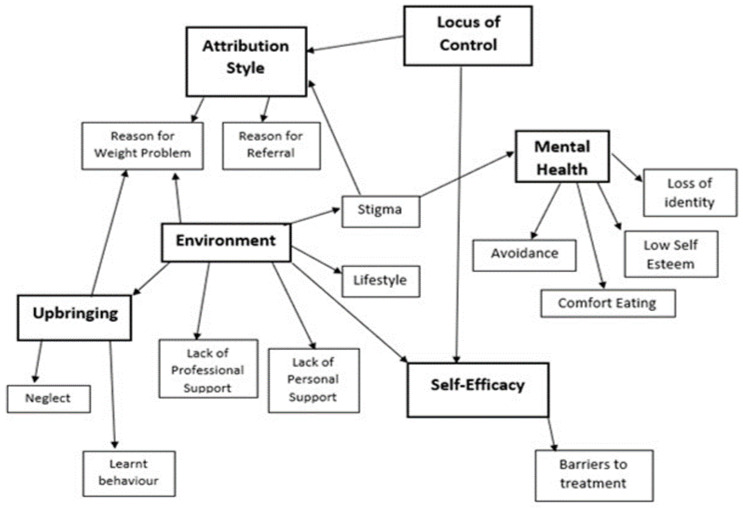
Final conceptual map of theoretical analysis.

**Figure 2 healthcare-12-00747-f002:**
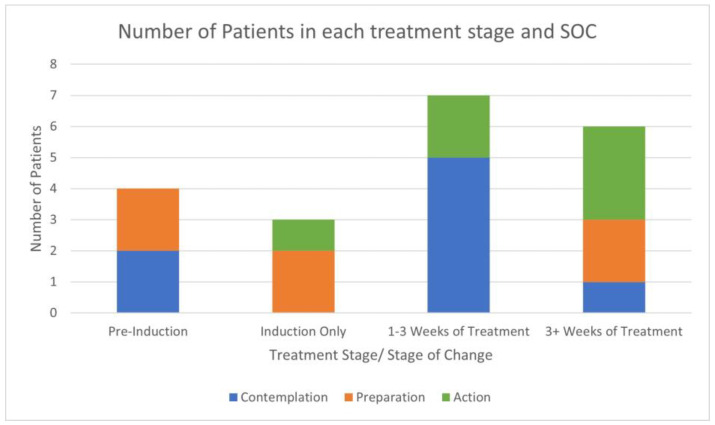
Bar graph showing number of participants by treatment, duration, and stage of change (SOC).

**Figure 3 healthcare-12-00747-f003:**
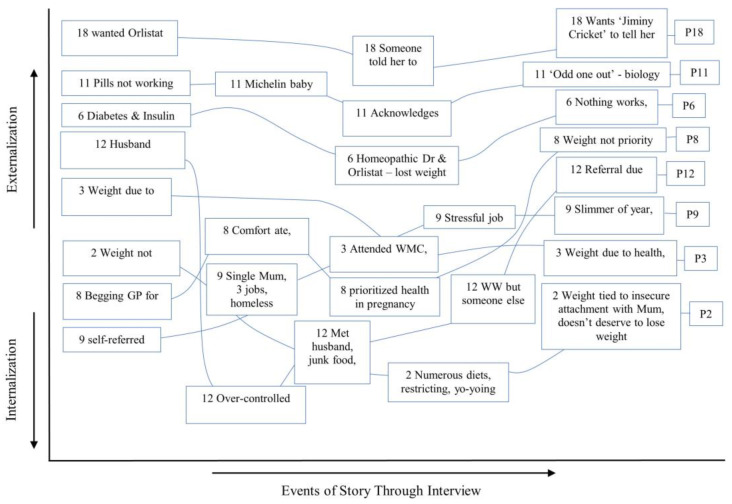
Storylines of participants in ‘Contemplation’.

**Figure 4 healthcare-12-00747-f004:**
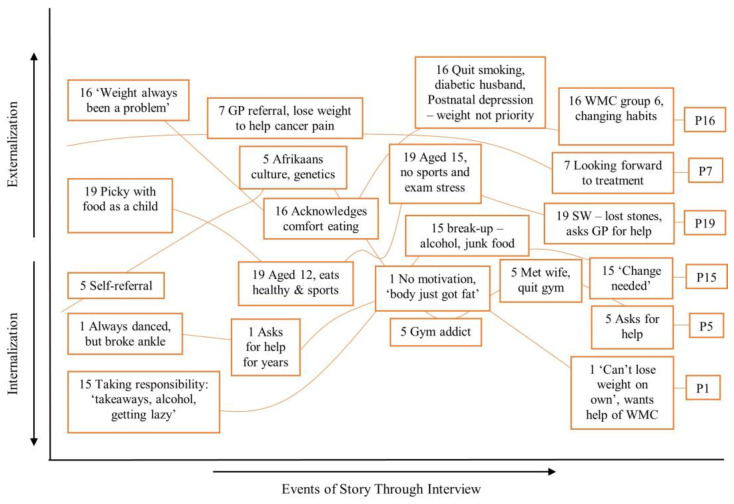
Storylines of participants in ‘Preparation’.

**Figure 5 healthcare-12-00747-f005:**
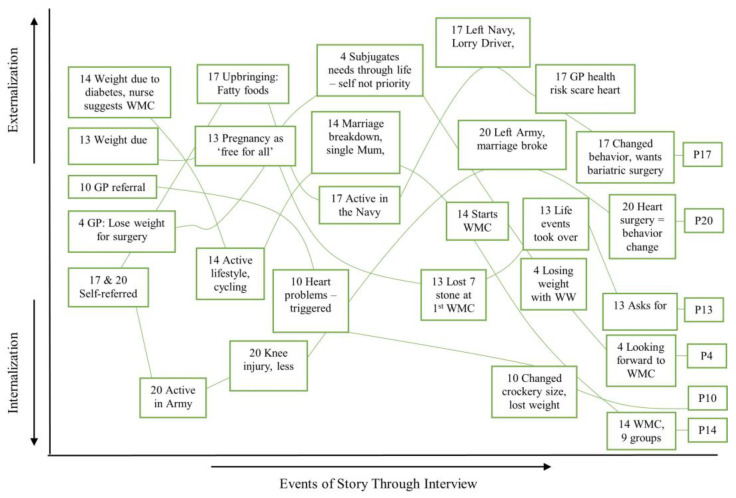
Storylines of participants in ‘Action’.

## Data Availability

The data that support the findings of this study are available upon request from the corresponding author. The data are not publicly available due to privacy or ethical restrictions.

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
