# Peer review of "Blending Behavioural Theory and Narrative Analysis to Explore the Lived Experience of Obesity and Assess Potential Engagement in a UK Weight Management Service: Theory and Narrative Approaches in Weight Management"

_healthcare, 2024, doi:10.3390/healthcare12070747_

Round 1
Reviewer 1 Report
Comments and Suggestions for Authors
An interesting study that presents the potential value of narrative based medicine within the context of weight management and obesity, and as a person centered approach to clinical treatment. The paper could be strengthened by addressing the points below:
Abstract: the first statement as Background is more of an opinion and needs to be supported by the literature. (Lines 13-14) Most recent and updated treatment guidelines for obesity - internationally- recognize a person centered, interdisciplinary approach as the gold standard to obesity treatment and clinical care.
Background/Introduction- recommend the authors provide a more updated and recent search of the literature. Many articles cited are very old (2003-2015) with the most recent articles being related to Covid (2020) and there is a 2019 paper cited. While it's understood that citing theories that are well-established will be older articles- treatment guidelines and best practices for obesity care have been updated and there are a lot of articles that can be cited within the last 5 years (which would be considered recent).
Materials and Methods: lines 18-19 Eligible participants were either waiting to start treatment at the WMC or were within the first nine weeks of treatment. Please provide clarification regarding 1. how long were participants waiting to start treatment? (weeks vs. months?) 2. how and why was it decided that eligibility would be for participants "within the first 9 weeks of treatment". This seems arbitrary and out of context. For example- how long was the treatment? What made a participant less eligible at 10 weeks of treatment?
It would also be helpful for the reader to understand more about the "WMC" from which these research participants were recruited. For example- how long was the treatment, was it interdisciplinary/who provided obesity treatment at this weight management clinic?
Discussion: Lines 335-338 don't seem to fit as an opening paragraph to this discussion and doesn't add to the discussion. Recommend starting the discussion with lines 339-340: This study explored whether patient narratives could be used to assess the utility of psychological theories of behavior in the context of motivation to engage with a WMC.
Lines 384 and 385: Society holds pervasive negative attitudes around obesity [46], where individuals are blamed and shamed for their weight [35]. The references cited here are very old (2009 and 2011). Recommend the authors find recent articles (in the past 5 years) that support this statement- which is a very broad statement.
The discussion might be strengthened by talking more about how while current treatment guidelines recognize that obesity treatment should be person centered and interdisciplinary, personal biases of healthcare providers may still exist. Perhaps citing literature regarding ways to address any biases, and to ensure person centered approaches are utilized in treatment would be more helpful in this discussion. Additionally, it is recommended that authors discuss how their data/results/approach for narrative medicine could positively contribute to obesity treatment- especially within the context of the psychological/social discipline that is recommended as part of interdisciplinary care for obesity/persons struggling with obesity.
Comments on the Quality of English LanguageMinor editing required.
Author Response
We thank the reviewer for taking the time to review our paper.
- See lines 13-14 – we have updated this statement and added updated references (guidelines/research) in relation to this (See lines 36-39)
- Materials and methods: More detail has been added directly addressing these points See lines 121-129
- Discussion: deleted erroneous intro and commenced at line 348
- Added “traditionally” to highlight that the point being made might not necessarily be current, but it is still valid in relation to our finding and previous literature
- Re discussion and the point raised about how we can apply our findings to the treatment of obesity, we specifically address this already in lines 398 to lines 419.
Reviewer 2 Report
Comments and Suggestions for Authors
This paper reports on a qualitative study of a small group of individuals living with obesity using a blending of specific psychological theories of behaviour with a narrative review of lived experience. While this is a valuable perspective there is no contextual literature referenced regarding psychological perspectives in obesity. This is not the first paper to try to use psychological theory in obesity. Nor is it the first paper to examine narrative approaches. I would like to see a more comprehensive review of the literature to justify this study.
We are told that the authors selected 4 theories from 82. How did they arrive at these theories? Why did they reject 78? Again, there is a extensive literature on integrating multiple behaviour theories together but there is no reference to work in this area, such as the theoretical domains framework. While I think this is an interesting study it would benefit from increased justification.
The sample for this study involves 20 people referred for weight management intervention. No mention of selection criteria and representativeness is included in the paper. We are told 20 consecutive patients were recruited. How many were approached, how representative are those who consented from those who didn't?
I wonder if the authors correctly identify the researchers in section 2.2. There is no mention of author 4 and author 2 is identified as having a lead role in the WMC and yet appears to be an MSc graduate student. I suspect the Endocrinologist (author 3) may be the lead clinician.
Very little details are given regarding the narrative interview. Were the topics listed in 2.3 addressed with all participants? Analysis of the transcripts was done using the constructs of attributional style, locus of control and self-efficacy. However we are not told how these constructs were evaluated. What categorization scheme was used and how was variability addressed? For instance, one might report different levels of self-efficacy for different behaviours. Was there are framework to address coding and consistency across behaviours, experiences, or relationships?
The results and discussion (note the authors copied the instructions for the discussion as their opening paragraph) are quite interesting but perhaps "over done". I am not sure that Figure 2 is really needed as the intervention itself would change how the person presented. I think it might be best for the authors to first present the categorization of stage, then examine AS, LoC and SE within these patterns. This would allow them to illustrate the value of the Labovian analysis.
Author Response
We thank the reviewer for their careful and thorough review. Our responses to the issues raised are detailed below in the order that they were presented.
- We have added more detail on psychological theories (see lines 54-75)
- Lines 80-108 provide an existing and clear rationale for why the theories were selected for analytical scrutiny. However to make it absolutely explicit we have added text (see lines 86-88)
- “The sample for this study involves 20 people referred for weight management intervention. No mention of selection criteria and representativeness is included in the paper”
- See lines 147-154 – we state this is a consecutive convenience sample and make no claims to representativeness
- We did not collect information on those who refused participation
- Correct authors identified see lines 141-146
- Very little details are given regarding the narrative interview. Were the topics listed in 2.3 addressed with all participants? Analysis of the transcripts was done using the constructs of attributional style, locus of control and self-efficacy. However we are not told how these constructs were evaluated. What categorization scheme was used and how was variability addressed? For instance, one might report different levels of self-efficacy for different behaviours. Was there are framework to address coding and consistency across behaviours, experiences, or relationships?
- Lines 164-181 describe how the analysis of transcripts generated and evaluated AS, LoC and SE. This explicitly states how we generated and evaluated these constructs using narrative analysis and visualisation through storylines. The framework was the narrative analysis – both structural and over narrative levels. We draw the reviewer’s attention to the table in appendix 1.
E.g. AUTHOR 1 read transcripts and crosschecked these with audio files. Transcripts coded into NVivo v12 software. A deductive theoretical analysis explored the text and utterances within each transcript and across all transcripts [30] to identify Attributional style (AS), Locus of Control (LoC), and Self-efficacy (SE) in relation to motivation to lose weight. When no new codes emerged, existing codes were collated for refinement and mapping to our theoretical framework. AUTHOR 1 applied Labov’s (1972) [31] structural analysis (see appendix Table A1, which explains Labovian codes) to investigate the patterning of stories, and salient content. Participants’ accounts of referral to the WMC were analysed as their ‘referral sub-plot’ and used to further scrutinize text related to AS and LoC. Transcripts were also analysed in their entirety to assess whether they contradicted or supported referral sub-plots. Participants were placed in a SoC category based on the theoretical and structural analyses, according to Norcross et al’s (2011) [18] definitions. Story lines were visualized to illustrate the shifts in externalization and internalization of LoC with the starting point being their referral sub-plot (see figures 3, 4, & 5).
Throughout the analysis we utilised Wong et al’s. (2018) [34] narrative analytic levels (personal, interpersonal and social). This provided a means to identify how stories were represented within the interviews, and the role of wider discourses on how the stories were told [35].
- The results and discussion (note the authors copied the instructions for the discussion as their opening paragraph) are quite interesting but perhaps "over done".I am not sure that Figure 2 is really needed as the intervention itself would change how the person presented. I think it might be best for the authors to first present the categorization of stage, then examine AS, LoC and SE within these patterns. This would allow them to illustrate the value of the Labovian analysis.
- We argue that the findings are robustly supported by the data, and strongly feel that figure 2 is important to present, because the intervention changes the way people present in line with AS, SE and LoC. The temporal nature of these changes is important to present.
- Secondly it is important to demonstrate evidential support for the constructs as viable for taking forward for narrative analysis, and also the reliability of these constructs across the sample. Hence ordering the findings as such. This also represents the stages of analysis in a transparent way and facilitates reproducibility.
Reviewer 3 Report
Comments and Suggestions for Authors
General comment and questions to authors:
·The study applied narrative interviews to explore the lived experience of individuals with obesity problem, including the causes of their weight gain and their expectation and engagement with treatment options. The authors have adequately described their methods/approaches and highlighted key findings. Their findings showed narrative approaches/story telling style interviewing helped both the researchers and the participants better understand the complexity of factors involved in weight gain, including internal and external ones, and how the person’s belief/locus of control determines the success of any treatment plan. The study highlights the need to stir away from reductionist approach to deal with obesity and focus a more tailored approach that acknowledges personal and contextual factors involved.
· However, this reviewer believes authors will improve the quality of the paper if they provide additional literature in the background section to highlight how the problem affects men and women (gender disaggregated information).
· The demographic composition of participants has quite a range (wide age range [from 20yrs -82yrs], all white, more than twice as many women than men) – Do authors think this has any implication on the nature of their finding and its interpretation (e.g., wider applicability?)
· Was there anything in your data to indicate men and women experience the challenge of obesity differently? Is compliance affected by gender? It would be great if authors elaborate on these aspects to improve the quality of their paper. When this is not possible, at least include some explanation in the limitation section.
Author Response
We thank the reviewer for taking the time to review our paper. The issues raised are dealt with in that order below:
- Provide additional literature in the background section to highlight how the problem affects men and women (gender disaggregated information)
- Please see additional text on lines 47-52.
- The demographic composition of participants has quite a range (wide age range [from 20yrs -82yrs], all white, more than twice as many women than men) – Do authors think this has any implication on the nature of their finding and its interpretation (e.g., wider applicability?)
- Almost certainly. Plymouth is a predominantly white British area and more women are diagnosed with obesity than men, they are also more likely to seek help for it. As such our sample is probably relatively speaking representative of the broader statistics, aside for issues of race and these have been added to in the limitations.
- Was there anything in your data to indicate men and women experience the challenge of obesity differently? Is compliance affected by gender? It would be great if authors elaborate on these aspects to improve the quality of their paper. When this is not possible, at least include some explanation in the limitation section.
- We present the results of the analysis as it surfaced and were mindful to illuminate gender differences where they were explicitly spoken about or interpretively surfaced. There were commonalities and divergences in experience across the sample, but these were not specifically told as related to gendered experiences.
- When the reviewer refers to “compliance” we can not comment since we did not probe this. This is because the theoretical framework from which we worked prioritised partnership and person centredness, therefore the term ‘compliance’ is not a salient construct within this paper.
Round 2
Reviewer 1 Report
Comments and Suggestions for Authors
Please see below for recommended revisions:
1. It is usually best to cite the specific article vs. use "ibid".
2. Please define "BPS" in line 70
3. Please define WMC in line 89
4. Please be sure your references are properly cited and your reference list is formatted according to the journal's preferred style/consistently
Comments on the Quality of English Languageminor edits for English
Author Response
Thank you. I have now corrected these errors on the updated MS.
Reviewer 2 Report
Comments and Suggestions for Authors
The authors have done a satisfactory job responding to reviewer's comments
Author Response
Thank you.